# Use of the Analytic Hierarchy Process and Selected Methods in the Managerial Decision-Making Process in the Context of Sustainable Development

**Jana Stofkova [1], Matej Krejnus [1], Katarina Repkova Stofkova [1,*], Peter Malega [2] and Vladimira Binasova [3]**

1   Department of Communication, University of Zilina, Univerzitna 1, 010 26 Zilina, Slovakia
2   Department of Industrial and Digital Engineering, Technical University of Kosice, Park Komenskeho 9, 040 02 Kosice, Slovakia
3   Department of Industrial Engineering, University of Zilina, Univerzitna 1, 010 26 Zilina, Slovakia
*   Correspondence: katarina.repkova@uniza.sk; Tel.: +421-917-144-757

**Abstract:** This article deals with the Analytic Hierarchy Process (AHP) method, which can be calculated in several ways. The aim of the paper is to analyze and describe the AHP method as essential for strategic managerial decision-making to determine which method is efficient for the calculation and to set the proper order of criteria. In the contribution, we show how the AHP method can be used through different techniques. In the article, there are included methods that can be used in order to calculate the matrix in the AHP process for setting criteria. This study also focused on the accuracy of various methods used to compute AHP. The paper contains the procedure of using the Saaty method through the Excel program. The results of the research show that the most accurate method is the Saaty method. In comparison with the Saaty method is the geometric mean method with the slightest deviation (CI = 0.00010), followed by the Row sum of the adjusted Saaty matrix with deviation (CI = 0.00256), reverse sums of the Saaty matrix columns (CI = 0.00852), Arithmetic mean and Row sums of the Saaty matrix (CI = 0.01261). All of these methods are easy to calculate and can be performed without major mathematical calculations. The AHP method is often used with other methods such as SWOT, FUZZY, etc. The survey was carried out through an inquiry with managers who graduated from universities in Slovakia and showed that the respondents considered the Saaty method as the most complex and the most difficult. The geometric mean and average mean methods were regarded as the simplest methods. Respondents (44%) stated that they were able to use a program to calculate the AHP. Respondents (46%) had experience with some method related to the strategic managerial decision-making process. Managers (72%) regarded this skill as important for decision-making in their managerial position. The contribution of this paper is to show the advantages of the AHP method in its wide use in various fields.

**Keywords:** AHP process; Saaty method; Excel program; managerial decision-making process

## 1. Introduction

The situation in world markets in companies or other institutions places high demands on the quality of managerial decision-making. An incorrect decision in any area can cause losses and sometimes even lead to the liquidation of businesses. Therefore, in preparing managers and other essential positions for the 21st century, it is crucial to return to the importance of mathematics and its applications, for example, in decision-making processes. This approach brings benefits to both smaller and larger companies. There are several methods such as AHP, e.g., TOPSIS, DEMATEL, ELECTRE, etc. The authors often use the AHP method and they consider this method as one of the most suitable for a lot of situations, and for this reason the paper is dedicated to this method. The AHP method is described in strategic managerial decision-making in detail and therefore the benefits of the AHP method and its wide use across different disciplines are discussed and highlighted.

The decision-making process is considered as one of the key aspects for entrepreneurs and managers. For quality decision-making, it is necessary to choose a suitable method which can be involved into decision-making. Also, the decision must be a logical process, which results in selecting the most appropriate alternative due to their significant impacts on the functioning of the part of the organization in which it takes place [1].

The AHP method has many applications in different fields and combinations of different methods. Therefore, it is helpful to understand several options for calculating the AHP, from more complex to more straightforward methods.

The article describes the procedure of using the Saaty method and compares the results with other methods. Therefore, this method is accessible even for inexperienced users concerning basic knowledge of mathematics to obtain comparable results, which is a great benefit. AHP can be used to determine the weight criteria and Grey Relational Analysis and TOPSIS is used to rank the bank performances [2,3].

The Saaty method can be used in managerial decision-making in several criteria and improves the analytical hierarchical process while helping to find the optimal alternative [4,5].

Professor Saaty described the method of multicriteria decision-making, which is now referred to by his name, although it is also known as the analytical hierarchical process (AHP) method [6].

In decision-making, people are usually influenced by their personality, environment, social and political background when deciding between options. It means that they make decisions based on their knowledge and experience, and that they analyze the risks and benefits of their decision. Decision-making can be freed from subjective influences by evaluating each alternative. It is easier to compare alternatives than to try to calculate their preferences. At the same time, the comparison has to be within the permissible range of consistency. The hierarchical analytical process (AHP) method includes both comparison and evaluation [6].

Multicriteria is a critical element of economic, social, political, military, and other decision-making. Decision-making processes are problem-solving processes with more than one possible solution. The process of solving a multicriteria decision-making task is a procedure by which the solver determines the optimal state of the system concerning more than one criterion. This procedure is called multicriteria optimization [3,5].

The Saaty method is used for the analysis and solution of decision-making tasks, with the help of which the solver selects the alternative that best meets the setting goal. The manager determines the alternatives and criteria using a pairwise comparison, compares the criteria and alternatives with each other and determines the preferences and the weight of the given preference [7–9].

The pairwise comparison method is also called the Saaty method. The mentioned method is used for multicriteria decision-making [10,11].

The analytical hierarchy process (AHP) method is accepted as the traditional method for determining the weighting of the suitability assessment. The authors used the AHP and coefficient of variation (AHP-CV) combined weight method to evaluate the suitability of urban green space better. Several authors use this method with another method to improve results. The research has shown that it can be extensively used in the related fields of planning concerning land use, green infrastructure, and transportation and tourism [12–14].

By promoting sustainable and transparent management, people in various economies could build and implement sustainable development practices and incorporate them into business activities. Sustainable development management integrates and connects relevant various sections of economics and science that have the most relevance for sustainable development. Current research problems solved by the AHP method is shown in Table 1.

**Table 1.** Current research problems solved by the AHP method.

| Reference | The Main Purpose of the Study | Criteria Considered | Methodology |
|---|---|---|---|
| Ranji, Parashkoohi and Zamani et al. (2022) [15] | Agricultural area | Agronomic, Technical, Economic, Environmental | AHP |
| F. Chan and H. Chan (2010) [16] | The selection of suppliers in the fast changing fashion market | Delivery, Quality, Assurance of Supply, Flexibility, Costs. | AHP |
| Kaymaz, Birinci and Kizilkan (2022) [17] | Sustainable development goals assessment of Erzurum province | Strengths, Weaknesses, Opportunities, Threats | AHP-SWOT |
| Imran, Agha, Ahmed et al. (2020) [18] | Simultaneous Customers and Supplier's Prioritization | Economic, Social, Environment | Fuzzy-AHP |
| Nikhil, Danumah, Saha et al. (2021) [19] | Forest Fire Risk Zone Mapping | Land cover types, slope angle, aspect, topographic wetness index, distance from the settlement, road, tourist spot | GIS and AHP |
| Ayyildiz, Gumus (2021) [20] | Hazardous material transportation: an application in Istanbul | Road, Environment, Traffic, Vehicle, Material | Pythagorean Fuzzy AHP |
| Ayyildiz, Gumus (2020) [21] | Petrol station location selection problem | Financial, Environmental, Opportunities, Supplier | Fuzzy AHP |
| Certain limitations were encountered during the preparation of the studies | A limited number of experts whose opinions were used to determine the priorities of the criteria determined by the AHP method. Experts avoided in-person meetings for binary comparisons in criteria priority determination due to the COVID-19 pandemic. These circumstances caused fewer consulted experts to be less than the desired number. The operation research models have some limitations using insufficient information, and it is challenging to incorporate the expert decision-making. Therefore, evaluation models are based on data analysis along with expert opinion. | | |

Source: own processing.

The most frequent experiments using the AHP method are related to semi-agriculture, forest fire risk zones, pumping station determination, and others. Other methods, such as GIS, Fuzzy or MAGDM, help solve common decision-making problems [19]. However, the researchers also encountered limitations, such as the limited number of experts whose views were used to prioritize the criteria. The AHP method is often used in combination with another method enabling its extension into different fields.

The authors used several methods of multicriteria processes. Among them, the analytic hierarchy process stands out as the most often used. AHP is used in power generation decision-making. The authors were able to select the best alternative using the AHP method [22,23]. The energy crisis in many countries has created pressure to decide the best alternative for generating energy and maintaining renewability. Renewability and environmental area is necessary to make the right decisions on the basis of the criterion. This creates room for the use of AHP to achieve the right approach [24,25].

## 2. Materials and Methods

The first method is a survey conducted among managers graduated from universities and colleges in the Slovak Republic. In addition to this method, the AHP process and various methods of its calculation were analyzed in order to find out the most accurate and the simplest method of calculation of the AHP process.

In addition to this, a survey was performed on a sample of 346 managers with the aim of detecting the managerial experience with mathematic methods supporting the decision-making in important issues. The survey was carried out by an electronic inquiry in the period of March to April 2022. All respondents were managers graduated from colleges or universities in Slovakia and worked at different levels of management, mainly primary and middle management, but also in senior management. More details are about the survey are listed in Table 2.

**Table 2.** Sample characteristics.

| Characteristics | Slovakia (N = 346) |
|---|---|
| Age (years) | |
| Means | 36.5 |
| Gender | |
| Male (%) | 61.6% |
| Female (%) | 38.4% |
| Graduation | |
| High school (%) | 41.1% |
| University (%) | 56.9% |
| PhD (%) | 2% |

Source: own processing.

Table 2 shows that men represented the largest group in the survey. In terms of education, the largest group of managers graduated from universities, followed by colleges and at least the PhD. graduated were involved in the survey. The results of this survey are discussed in the section "Results of the survey".

Article selection procedure:

(i)  The literature contains a scope of use of the AHP method in various areas. Many authors have used the AHP method in the decision-making process to select the best alternative from several options. The literature provides an overview of the use of the AHP method, but not many articles address the accuracy or managers' experience in terms of the final decision. There is no evidence of how to calculate the AHP method using software in the literature so that it is available even for start-ups and is free for use. According to the literature the AHP method is often used with other methods such as SWOT, FUZZY, etc.

(ii)  There are research gaps in how managers perceive the AHP method or another method in the decision-making process. Is it possible to use the AHP method in the EXCEL environment? What would enable the use of the method for start-up businesses and managers? Which calculation is appropriate to choose when calculating AHP in order to achieve the best possible result? What barriers does AHP have in its use from the research carried out so far?

(iii)  The article includes several calculation methods for verifying accuracy and determining of the most accurate method. The results were verified on two matrices to find out if the results were the same and whether there were some deviations in the calculation which was not confirmed. EXCEL software was selected for use to determine the difficulty of the calculation. The barriers to the use of the AHP method were identified based on the selected articles. In order to find out the attitude of managers, a questionnaire survey was carried out in the Slovak republic and the results will be implemented in managerial decision-making.

(iv)  Is it possible to use another AHP method in the managerial decision-making process that is available to the general public? The realization of calculations of various methods in order to verify the size of the deviations and determine their difficulty during the survey. In order to achieve relative verification, we calculated two matrices. Is it possible to use the AHP method and all its calculation methods in the EXCEL program? The calculations can be carried out in the selected software in order to verify whether all methods can be calculated in EXCEL. Managers' attitude towards the AHP method in the decision-making process after determining its difficulty. In order to obtain the opinion of managers on the AHP method, we conducted a survey that provided answers to our research questions (which are listed and already answered in the last table of the contribution). How does one identify barriers to the use of AHP?

A suitable solution is to review the implemented objectionable articles, supplementing them with the experience of the authors of the article.

### 2.1. AHP Process

The AHP method follows three steps in the decision-making process, including (1) the structure of the model; (2) the comparison of the criteria and alternatives and calculation of the weights; and (3) the synthesis of priorities. The basic feature of the AHP method is the representation of the complete decision-making issue as a specific hierarchical structure. Within the hierarchical structure, we distinguish the tree view with several levels representing the individual parts of the decision-making process. Each of them includes several elements. The top level of the hierarchy always contains only one element, which is the objective of the evaluation. A standard example of the AHP is a three-level hierarchy; for a more comprehensive picture, they can also be assigned under criteria. The method is based on the assumption of evaluation consistency; it also arises from the hypothesis that inconsistency occurs mainly in evaluations between alternatives of seemingly minor importance to the decision making manager [19].

### 2.2. Hierarchical Structure

Hierarchical structures can be divided into two types. with criteria and alternatives and with criteria below the criteria and alternatives. Subsequently, it is necessary to recalculate the individual weights between the sub-criteria. The structure of the AHP process is displayed in Figure 1.

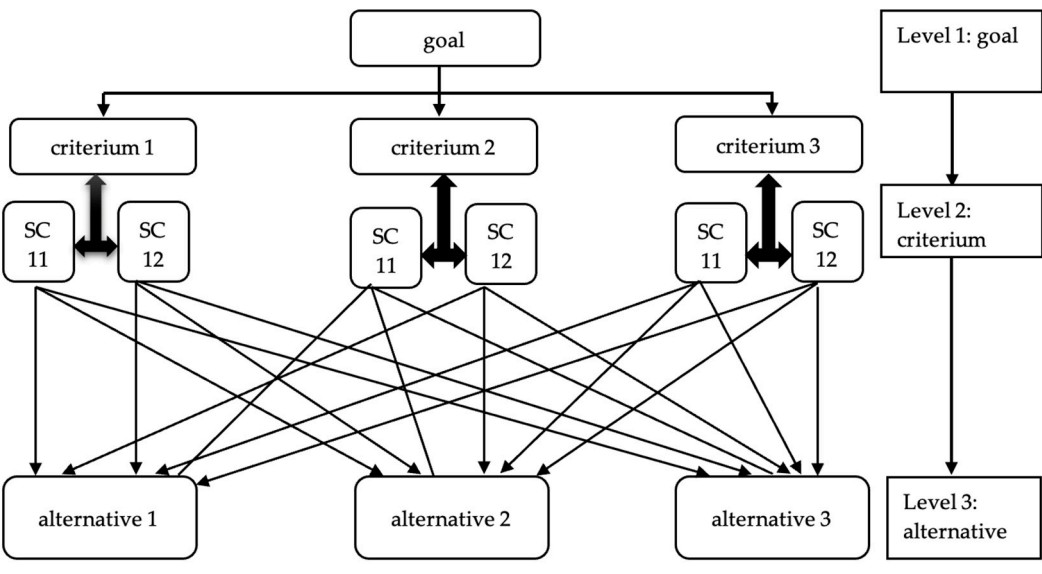

**Figure 1.** The structure of the AHP method.

The level 1 is the goal of the analysis. The level 2 is a multi-criterion consisting of several criteria or sub-criteria. PK—sub-criterion.

Level 1—the goal of the analysis—arrangement of alternatives.

Level 2—evaluation criteria: evaluation of the importance of the criteria.

Level 3—assessment of alternatives evaluated by the importance of alternatives.

The first step in the AHP process is a pairwise comparison between each criterion and possibly sub-criterion. Next, the individual criteria have to be assessed based on the scales enlisted in the following table. Table 3 shows the determination of the evaluation of the individual criteria with the preference rate from which the criteria are selected for the construction of the matrix. An example of comparison scales is listed in Table 3.

**Table 3.** The Saaty scale.

| Scales | Degree of Preference |
|---|---|
| 1 | The criteria are equally important |
| 3 | Medium importance of one factor over another |
| 5 | Strong or essential |
| 7 | Crucial importance |
| 9 | Extremely important |
| 2, 4, 6, 8 | Intermediate values |

Source: own processing.

The comparison results for each pair of factors were described as an integer value from 1 (the same importance) to 9 (extremely different importance), where a higher number means that the selected factor is more important than the other compared factor. Table 4 can help identify individual scales. The markings in Table 4 explain the assignment of values in the A matrix.

$$
(A) = \begin{matrix} & X1 & X2 & X3 & \ldots & Xn \\ X1 & 1 & \mathbf{4} & \mathbf{5} & \ldots & 5 \\ X2 & 1/4 & 1 & \mathbf{1/2} & \ldots & 3 \\ X3 & 1/5 & 2 & 1 & \ldots & 7 \\ \ldots & \ldots & \ldots & \ldots & 1 & \ldots \\ Xn & 1/5 & 1/3 & 1/7 & \ldots & 1 \end{matrix}
$$

**Table 4.** Choosing the values in the Saaty scale.

| Factor | Factor Weighting Score | | | | | | | | | | | | | | | | | Factor |
|---|---|---|---|---|---|---|---|---|---|---|---|---|---|---|---|---|---|---|
| | More Important | | | | | | | | Equally Important | Less Important | | | | | | | | |
| X1 | 9 | 8 | 7 | 6 | 5 | 4 | 3 | 2 | 1 | 2 | 3 | 4 | 5 | 6 | 7 | 8 | 9 | X1 |
| X2 | 9 | 8 | 7 | 6 | 5 | 4 | 3 | 2 | 1 | 2 | 3 | 4 | 5 | 6 | 7 | 8 | 9 | X2 |
| X3 | 9 | 8 | 7 | 6 | 5 | 4 | 3 | 2 | 1 | 2 | 3 | 4 | 5 | 6 | 7 | 8 | 9 | X3 |

Source: own processing.

After determining the individual weights, it is necessary to create a matrix. The matrix (A) shows the breakdown of the individual weights.

*2.3. Consistency Index*

Before calculating, it is necessary to check the table to ensure that it is sufficiently consistent and that there are no discrepancies within the pairwise comparison of the individual criteria. The consistency can be explained in the following comparison. If criterion X2 is three times more important than criterion X3 and X1 is three times more important than X2, then X1 is six times more significant than X3. In this case, the matrix is consistent. Figures 2 and 3 show the mentioned principle in visual form. The figure on the left shows the matrix consistency and the figure on the right shows the matrix inconsistency.

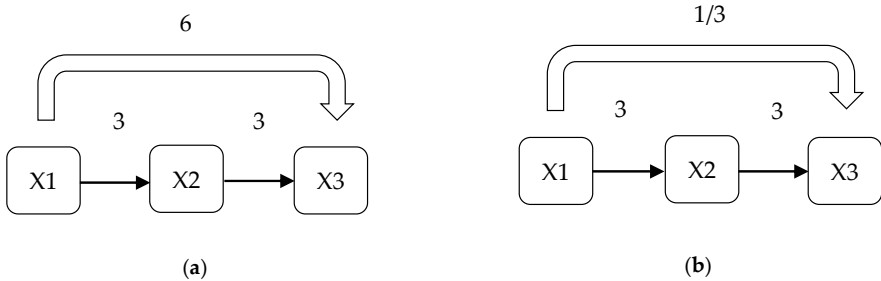

(**a**)  (**b**)

**Figure 2.** Principle of matrix consistency and inconsistency. (**a**) The matrix is consistent. (**b**) The matrix is inconsistent.

| | A | B | C | D | E | F | G | H |
|---|---|---|---|---|---|---|---|---|
| 102 | | **Eigenvector** | | | | | | |
| 103 | | **A** | **B** | **C** | **D** | **E** | | |
| 104 | **A** | 1 | 2 | 4 | 1/2 | 1/3 | | |
| 105 | **B** | 1/2 | 1 | 3 | 1/4 | 1/6 | | MatrixA |
| 106 | **C** | 1/4 | 1/3 | 1 | 1/5 | 1/8 | ← | |
| 107 | **D** | 2 | 4 | 5 | 1 | 1 | | |
| 108 | **E** | 3 | 6 | 8 | 1 | 1 | | |
| 109 | | | | | | | | |
| 110 | | **A** | **B** | **C** | **D** | **E** | | |
| 111 | **A** | 1 | 0 | 0 | 0 | 0 | | |
| 112 | **B** | 0 | 1 | 0 | 0 | 0 | | MatrixB |
| 113 | **C** | 0 | 0 | 1 | 0 | 0 | ← | |
| 114 | **D** | 0 | 0 | 0 | 1 | 0 | | |
| 115 | **E** | 0 | 0 | 0 | 0 | 1 | | |

**Figure 3.** Matrix A and Matrix B.

In Figure 3, a blue arrow is shown, which determines the name (designation) of the matrix, e.g., matrix A, matrix B.

If X1 is only 1/3 more significant than X3 then the matrix will not be consistent. We verify the consistency using the matrix consistency index and calculate it according to the formula.

$$CI = \frac{\lambda_{max} - x}{x - 1} \tag{1}$$

$$CR = \frac{CI}{RI} \tag{2}$$

CI—consistency index; Saaty matrix is sufficiently consistent if CI < 0.1
X—is the number of criteria.
$\lambda_{max}$—specifies the largest correct number of the matrix.
CR—consistency ratio, consistency value.
RI—table value based on the number of criteria, so-called random index.

If the calculation included multiple criteria with multiple sub-criteria, the so-called composite score would be used.

A composite score is calculated. First, we compute the local weights of the criterion and the local weights of the parent criterion. We then multiply these values to obtain the global weights. We use the computed values from the Saaty matrix in the matrix to convert the input values. Once the values are normalized, we use one of the AHP calculation methods. In the last step, the obtained values are then added to the next global weights. We only add the values calculated in this way to obtain the composite score. The best choice will then be the largest value and the worst will be the smallest value.

*2.4. Methods of Calculation of the Saaty Method*

The Saaty method: eigenvalue and eigenvector. For comparison and reliability of individual calculation methods, calculations will be performed on one type of example by all selected methods. We can divide the individual calculations on the Saaty procedure, which is implemented using eigenvector and eigenvalue, i.e., custom vector matrix. The result must be standardized. The left-hand side represents matrix-vector multiplication, but the right-hand side is scalar-vector multiplication. Firstly, we can rewrite that right-hand side as some kind of matrix-vector multiplication using a matrix which has the effect of scaling any vector by a factor of λ. Using the unit matrix, we obtain the final Formula (4).

$$A * V = \lambda_{max} * v \tag{3}$$

$$\begin{bmatrix} \lambda & 0 & 0 \\ 0 & \lambda & 0 \\ 0 & 0 & \lambda \end{bmatrix} \quad \lambda * \underbrace{\begin{bmatrix} 1 & 0 & 0 \\ 0 & 1 & 0 \\ 0 & 0 & 1 \end{bmatrix}}_{I}$$

$$(A - \lambda_{max} * I) * v = 0 \tag{4}$$

v—custom vector, so-called eigenvector
$\lambda_{max}$—the corresponding eigenvalue
$v * \lambda_{max}$—vector scale
A—decision matrix
I—unit matrix

This method serves as a basis for comparison with other calculation methods and for determining the accuracy of other methods. Of all these approaches, this is the most complex and the most suitable one. Due to its complexity, it is used in specialized decision support programs such as Super decision, Expert choice Comparison and the Priority Estimation tool.

The following table summarizes the individual calculation methods and the equations that will be implemented. In addition, there are several methods for calculating the matrix, which are demonstrated in Table 5. In other words, Table 5 shows the methods for calculating in different ways.

**Table 5.** Methods for calculating the Saaty decision matrix.

| Method Description | Geometric Mean | Arithmetic Mean | Row Sum of the Adjusted Saaty Matrix | Reverse Sums of Saat Matrix Columns | Row Sums of the Saaty Matrix |
|---|---|---|---|---|---|
| Method of calculation | The method is based on calculating the geometric mean of the individual rows of the decision matrix. | This is the simplest method of calculation. It consists of averaging the matrix's individual rows that need to be standardized. | The method consists of two steps. The first step is to modify the decision matrix by dividing each column by the sum of the columns. In the second step, a row wise summation is performed. | The method is based on the principle that the preference intensity vector is calculated as the inverse of the columns of the matrix. | The method of calculation is based on a simple sum of the rows of the decision matrix. |
| Equation | $v_{i'} = \sqrt[x]{\prod s_{ij}}$ | $v_{i'} = \sum \frac{s_i}{x}$ | $v_{i'} = \sum \frac{s_{ij}}{T_j}$ | $v_{i'} = \frac{1}{\sum s_{ij}}$ | $v_{i'} = \sum s_{ij}$ |
| The result must be standardized | yes | yes | yes | yes | yes |
| Explanation of acronyms | $v_i$—unstandardized vector of preference intensities $S_{ij}$—elements of the decision matrix $T_j$—sum of elements of the j-th column | | | | |

Source: own processing.

The calculations will be performed by the Saaty method and compiled with the following methods: the row sum of the adjusted Saaty matrix, the geometric mean, the arithmetic mean, inverted sums of the columns of the Saaty matrix and the row sum of the Saaty matrix. Standardized results are necessary to gain the correct result.

*2.5. Evaluation of Alternatives Is Discussed Below*

To evaluate the criteria and partial alternatives, the overall evaluation of alternatives $H^j$ from another method are calculated.

$$H^j = \sum_{i=1}^{n} v_i * h_i^j \tag{5}$$

$H^j$—overall rating of the alternatives.

$h_i^j$—partial rating of the j-th alternative in relation to the i-th criterion.

$v_i$—criteria weights

A higher value determines the better fulfillment of the target $H^j$. For example, a value after standardization of 0.543 better meets our requirement than a value of 0.034, i.e., 0.543 > 0.034. The optimal alternative is the one with the highest overall rating.

The values for setting the criteria were determined based on the measurement carried out for selecting routes for the carrier. Criteria A, B, C, D, and E represent time, route length, costs, type of transport and type of route.

The determination of the weights was carried out according to academic experience at the university, and the criteria are used in the calculations and serve as the basis for the Saaty matrix.

## 3. Results

The comparison of the calculation methods for the AHP process is one of the main goals of this paper. An example of the matrix for calculations with the selected methods is shown in Table 3. The matrix contains five criteria from A to E. The matrix has a size of 5 × 5 elements. In order to compare it with other methods, it is necessary to perform the calculation by the Saaty method. Matrix (A) shows the criteria embedded in the matrix.

$$(A) = \begin{array}{c} \\ A \\ B \\ C \\ D \\ E \end{array} \begin{array}{ccccc} A & B & C & D & E \\ \left( \begin{array}{ccccc} 1 & 2 & 4 & 1/2 & 1/3 \\ 1/2 & 1 & 3 & 1/4 & 1/6 \\ 1/4 & 1/3 & 1 & 1/5 & 1/8 \\ 2 & 4 & 5 & 1 & 1 \\ 3 & 6 & 8 & 1 & 1 \end{array} \right) \end{array}$$

### 3.1. The Saaty Method

The calculation will be performed based on (3) $A*v = \lambda_{max}*v$. The Saaty procedure was calculated using Excel and its functions. To the created symmetric Matrix A it is necessary to design a single-unit Matrix B. Figure 3 shows Matrix A and Matrix B, which are used for calculation in the next steps.

In order to obtain the lambda sign, it is necessary to determine the first estimation of the lambda, which can be any number, e.g., 999. Then we use Formula (4) and multiply the matrixes with the estimated lambda. Based on this matrix, we compute the determinant of the matrix using the MDETERM function. From the menu of MS Excel, one shall click Tools-Goal Seek and the Goal Seek dialog window shows up. After (set cell: determinant; to value = 0; by changing cell: $\lambda_{max}$). The lambda matrix is 5.102357 (see Figure 4). The blue arrow indicates the formula that was used for the calculation.

**Figure 4.** Calculation based on Formula (4).

The consistency matrix is calculated using relations (1) and (2) and reported in Table 4. To calculate the matrix vectors, we use the matrix from Figure 5 and the Sumproduct function shown in Figure 5. Diagonally, the numbers are highlighted with blue undertones, as we leave the unit matrix.

| B119 | | fx {=MetrixA-$H$123*MetrixB} | | | | | | |

| | | A | B | C | D | E | (function) Sumproduct | |
|---|---|---|---|---|---|---|---|---|
| 125 | | | | | | | | |
| 126 | A | -4.102 | 2.000 | 4.000 | 0.500 | 0.333 | 15.56 | |
| 127 | B | 0.500 | -4.102 | 3.000 | 0.250 | 0.167 | 3.13 | |
| 128 | C | 0.250 | 0.333 | -4.102 | 0.200 | 0.125 | -9.97 | |
| 129 | D | 2.000 | 4.000 | 5.000 | -4.102 | 1.000 | 13.59 | |
| 130 | E | 3.000 | 6.000 | 8.000 | 1.000 | -4.102 | 22.49 | |
| 131 | Vectors | 1.000 | 2.000 | 3.000 | 4.000 | 5.000 | Sum= | 15.000 |
| 132 | Normalization | 0.06667 | 0.13333 | 0.200000 | 0.26667 | 0.33333 | Sum= | 1 |

**Figure 5.** Principle of matrix consistency and inconsistency.

It is unnecessary to choose random vectors, for example 1, 2, 3, 4, 5. To calculate the matrix vectors, a manager uses the matrix from Figure 5 and clicks the Sumproduct function, as shown in Figure 6. Vectors can be selected in any value, for example 1, 2, 3, 4, 5. From the MS Excel menu, one shall click "solver parameters" and choose the values as shown in Figure 6.

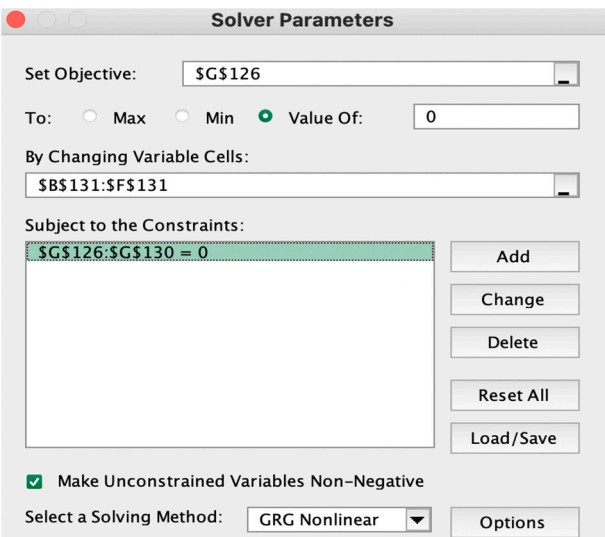

**Figure 6.** Solver function in Excel program.

Based on the results of the solver parameters and in order to calculate the vectors of the matrix, it is important to use the matrix $(A - \lambda_{max}*I)*v = 0$ and use the Sumproduct function, as shown in Figure 6. It is unnecessary to choose random vectors determined in Figure 5. From the MS Excel menu, one shall click "solver parameters" and choose the values as shown in Figure 6. Finally, the results of the vector are unstandardized, and it is necessary to standardize the vector, such as (1.932/12.411 = 0.15570), etc. Figure 7 shows vector values after standardization v = (0.15570; 0.08646; 0.04397; 0.31181; 0.40207). Table 6 shows the truncated matrix calculation results by means of the Saaty method. Diagonally, the numbers are highlighted with blue undertones, as we leave the unit matrix.

| G126 | | fx =SUMPRODUCT(B126:F126;$B$131:$F$131) | | | | | | |

| | | A | B | C | D | E | (function) Sumproduct | |
|---|---|---|---|---|---|---|---|---|
| 125 | | | | | | | | |
| 126 | A | -4.102 | 2.000 | 4.000 | 0.500 | 0.333 | 0.00 | |
| 127 | B | 0.500 | -4.102 | 3.000 | 0.250 | 0.167 | 0.00 | |
| 128 | C | 0.250 | 0.333 | -4.102 | 0.200 | 0.125 | 0.00 | |
| 129 | D | 2.000 | 4.000 | 5.000 | -4.102 | 1.000 | 0.00 | |
| 130 | E | 3.000 | 6.000 | 8.000 | 1.000 | -4.102 | 0.00 | |
| 131 | Vectors | 1.932 | 1.073 | 0.546 | 3.870 | 4.990 | Sum= | 12.411 |
| 132 | Normalization | 0.15570 | 0.08646 | 0.04397 | 0.31181 | 0.40207 | Sum= | 1 |

**Figure 7.** The results and standardization of vectors.

**Table 6.** Results of the Saaty method.

| Alternative | Values | Order | |
|---|---|---|---|
| A | 0.15570 | 3 | |
| B | 0.08646 | 4 | $\lambda_{max}$ = 5.10235 |
| C | 0.04397 | 5 | N = 5 |
| D | 0.31181 | 2 | RI = 1.21 |
| E | 0.40207 | 1 | CI = 0.02110 |

Source: own processing.

The value of the consistency ratio CR is less than 0.10, which is well within the acceptable range. Tables 7 and 8 below summarize all the results. In particular, Table 7 shows the results of the other methods.

**Table 7.** Results after calculation and ranking.

| Methods | Geometric Mean | Arithmetic Means | Row Sum of the Adjusted Saaty Matrix | Reverse Sums of Saaty Matrix Columns | Row Sums of Saaty Matrix |
|---|---|---|---|---|---|
| Used formula | $v_{i'} = \sqrt[x]{\prod s_{ij}}$ | $v_{i'} = \sum \frac{s_i}{x}$ | $v_{i'} = \sum \frac{s_{ij}}{T_j}$ | $v_{i'} = \frac{1}{\sum s_{ij}}$ | $v_{i'} = \sum s_{ij}$ |
| The result has to be standardized | yes | yes | yes | yes | yes |

Source: own processing.

**Table 8.** Results of calculations.

| Methods | Geometric Mean | Arithmetic Means | Row Sum of the Adjusted Saaty Matrix | Reverse Sums of Saaty Matrix Columns | Reverse Sums of Saaty Matrix | Ranking |
|---|---|---|---|---|---|---|
| Altern-atives | | | Values | | | - |
| A | 0.15768 | 0.16789 | 0.16453 | 0.14954 | 0.16789 | 3 |
| B | 0.08550 | 0.10538 | 0.09417 | 0.07570 | 0.10538 | 4 |
| C | 0.04330 | 0.04090 | 0.04436 | 0.04807 | 0.04090 | 5 |
| D | 0.31131 | 0.27862 | 0.29334 | 0.34216 | 0.27862 | 2 |
| E | 0.40221 | 0.40722 | 0.40359 | 0.38453 | 0.40722 | 1 |
| $\lambda_{max}$ | 5.10190 | 5.16337 | 5.16337 | 5.14357 | 5.16337 | - |
| CI | 0.02105 | 0.03375 | 0.02371 | 0.02966 | 0.03375 | - |

Source: own processing.

The results were standardized to obtain relevant data for comparison. The results show that the ranking of the individual alternatives is very similar, provided that the matrix is sufficiently consistent under the assumption that we can achieve the same ranking of alternatives. Furthermore, we can detect deviations in the values between the individual types of calculation. The arithmetic mean and the row sum method of the Saaty matrix display the same results.

From the table mentioned above, we can see the differences in the individual calculations (Table 9). The results showed the Saaty method to be the most accurate, followed

by the geometric mean method (see Table 10), where we noticed the slightest deviation. On the other hand, the exact methods are Row sum of the adjusted the Saaty matrix and Arithmetic mean. However, these methods are the easiest to be calculated and can be performed without major mathematical calculations. Deviations occurred in all methods, but the most accurate method is the Saaty method, and accurate results were also obtained by using the geometric mean.

**Table 9.** Comparison of all methods with the Saaty method.

| | Method | Geometric Mean | Arithmetic Mean | Row Sum of the Adjusted Saaty Matrix | Reverse Sums of Saaty Matrix Columns | Row Sums of The Saaty Matrix |
|---|---|---|---|---|---|---|
| Deviation from Saaty method | Average of calculation results | 0.00103 | 0.01684 | 0.00885 | 0.01284 | 0.01684 |
| | Maximum | 0.00198 | 0.03319 | 0.01846 | 0.03036 | 0.03319 |
| | Minimum | 0.00050 | 0.00307 | 0.00040 | 0.00410 | 0.00307 |
| | $\lambda_{max}$ | 0.00046 | 0.06101 | 0.01238 | 0.04122 | 0.06101 |
| | CI | 0.00010 | 0.01261 | 0.00256 | 0.00852 | 0.01261 |

Source: own processing.

**Table 10.** Ranking of methods according to their order based on results.

| | Methods | Order | Equations |
|---|---|---|---|
| Ranking based on the accuracy of the method | Saaty method | - | $(A - \lambda_{max} * I) * v = 0$ |
| | Geometric mean | 1 | $v_{i'} = \sqrt[x]{\prod s_{ij}}$ |
| | Row sum of the adjusted Saaty matrix | 2 | $v_{i'} = \sum \frac{s_{ij}}{T_j}$ |
| | Reverse sums of the Saaty matrix columns | 3 | $v_{i'} = \frac{1}{\sum s_{ij}}$ |
| | Arithmetic mean | 4 | $v_{i'} = \sum \frac{s_i}{x}$ |
| | Row sums of the Saaty matrix | | $v_{i'} = \sum s_{ij}$ |

Source: own processing.

To verify the correctness of the calculations and the order of the individual methods, we calculated the random matrix (B).

$$
(B) = \begin{matrix} & \begin{matrix} A & B & C & D & E \end{matrix} \\ \begin{matrix} A \\ B \\ C \\ D \\ E \end{matrix} & \begin{pmatrix} 1 & 3 & 6 & 7 & 6 \\ 1/3 & 1 & 3 & 4 & 2 \\ 1/6 & 1/3 & 1 & 4 & 2 \\ 1/7 & 1/4 & 1/4 & 1 & 2 \\ 1/6 & 1/2 & 1/2 & 1/2 & 1 \end{pmatrix} \end{matrix}
$$

From the calculation results, we can detect the same result as for matrix (A), implying that the results are equal even when repeatedly calculated with a different matrix. Values of matrix B with Saaty method: where (A = 0.52081 B = 0.21760 C = 0.12776 D = 0.06675 E = 0.06704) $\lambda_{max}$ = 5.39447 CI = 0.08150. The results of the other methods are shown in Table 11.

**Table 11.** Matrix B control statement.

| Methods | Geometric Mean | Arithmetic Mean | Row Sum of the Adjusted Saaty Matrix | Reverse Sums of Saaty Matrix Columns | Row Sums of the Saaty Matrix | Ranking |
|---|---|---|---|---|---|---|
| Alternatives | | | | Values | | - |
| A | 0.53494 | 0.48791 | 0.53135 | 0.56383 | 0.48791 | 1 |
| B | 0.21495 | 0.21913 | 0.22517 | 0.20089 | 0.21913 | 2 |
| C | 0.12106 | 0.15910 | 0.12358 | 0.09493 | 0.15910 | 3 |
| D | 0.06327 | 0.07722 | 0.05261 | 0.06185 | 0.07722 | 5 |
| E | 0.06578 | 0.05664 | 0.06729 | 0.07850 | 0.05664 | 4 |
| $\lambda_{max}$ | 5.39038 | 5.52760 | 5.43108 | 5.50335 | 5.52760 | - |
| CI | 0.09759 | 0.13190 | 0.10777 | 0.12584 | 0.13190 | - |

Source: own processing.

### 3.2. The Results of the Survey among Managers

The survey performed among managers on different levels of management showed that the most complex method of the AHP process in managerial decision-making is the Saaty method (see Figure 8). Most managers regarded this method as a more difficult way of calculation compared with other methods.

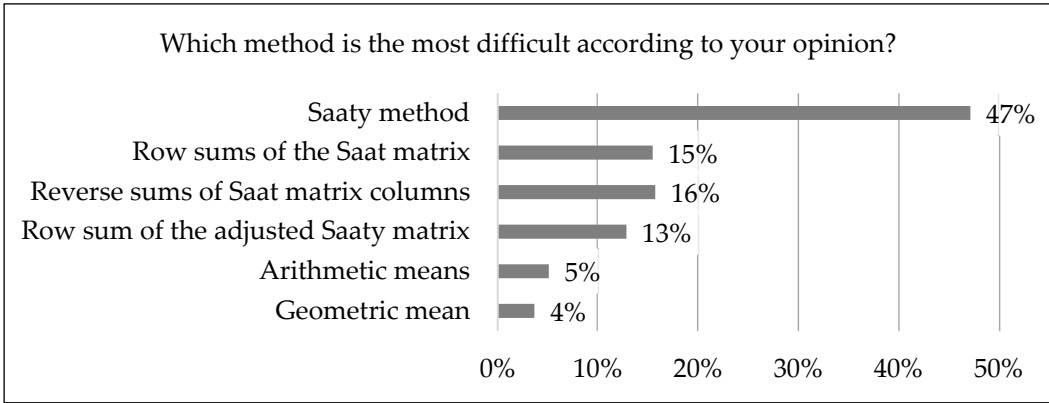

**Figure 8.** The difficulty requirements of the studied methods within the AHP process from the point of view of managers.

Therefore, it can be stated that the other methods are less demanding. Managers considered the arithmetic and the geometric mean as the least difficult. Sixty-nine percent of the respondents marked the "definitely no" option because the complexity of the Saaty method increases with more alternatives and the other methods give almost the same result and a simpler method of calculation at the same time. Furthermore, the respondents only emphasize the difficulty of the method when calculating without specialized software or knowledge of Excel calculation methods. Further questions of the survey concerning the decision-making process among managers are shown in Table 12.

**Table 12.** Further questions of the survey concerning the decision-making process among managers.

|   | Questions of the Survey | Definitely Yes | Rather Yes | Maybe | Rather No | Definitely No |
|---|---|---|---|---|---|---|
| 1 | Is the Saaty method your choice when the same outputs are achieved? | 15% | 4% | 2% | 9% | 69% |
| 2 | Could you use some program to calculate the Saaty method? | 35% | 9% | 29% | 18% | 9% |
| 3 | Do you have some experience with any method for decision making? | 46% | - | 18% | - | 35% |
| 4 | Do you think that it is important to know and control/understand, use a method for decision making in management positions? | 72% | 16% | 5% | 3% | 4% |

Source: own processing.

### 3.3. Evaluation of the Results

In the first question, most respondents (69%) marked the option "definitely no". According to managers, if the result of the Saaty method achieves the correct calculation and ranking of a specific method, the selected method of calculation is not important. In the second question, the respondents (44%) answered that they were capable of using a program to calculate the AHP. There are several software products, but their main limitation is price or subscription. In the third question, respondents (46%) stated they had experience with some method related to the managerial decision-making process. In the fourth question, the respondents (72%) regarded it as important to manage some methods for decision-making in managerial positions.

From the survey results, we can conclude that the outcomes confirmed our assumptions when using the specific calculating methods. Certain methods are simpler, and this is especially appreciated by college and university graduates who do not have experience with more complex types of calculations or software.

In order to assess which procedure is the most accurate, it was necessary to calculate the matrix using the Saaty method, or eigenvalue. Moreover, we tested all the procedures that could be used to calculate the matrix and determine the order of the criteria. The results showed that the second most accurate calculation method is the geometric mean method. A positive result is that all the procedures showed the same order of criteria. Furthermore, this fact opens the question for further research in terms of how many criteria are necessary for the different calculation methods to show different rankings. From the results of scientific works, we can conclude that the AHP method has a wide application in various fields, from IT to management.

The decision-making via the AHP method, for instance personnel department, could take the decide on selecting the applicants, because it is necessary to determine the criteria of the applicants in order to select the most appropriate one. The method must be combined with human decision. The next example, where it can be used, is in the logistic department, when a manager needs to select the best supplier for the company. The criteria can be the size of the company, the number of trucks, and the use of Just in Time, etc. The AHP can be used wherever a high-quality decision based on criteria is necessary.

## 4. Discussion

Another positive finding is that even a person with a lesser knowledge of mathematics can use the AHP method to achieve the same order of elements as in the most demanding method, which is the Saaty method.

In addition, this document provides a complete framework and all calculation methods for setting the criteria in the AHP method. The reader can choose which method of

calculation suits him, based on his mathematic skills and his work goals to achieve the desired result.

Decision-making is a key element for all businesses. Decision-making is essential for companies to make a good decision for leading the company in the present and future. The results from the AHP method can improve the company's performance.

Certain limitations were encountered during the preparation of the studies, such as the limited number of experts to determine individual weights for selected areas, which, however, depends on the area addressed. In addition to these limitations, for many alternatives it is recommended to use software that calculates many alternatives.

With sufficient knowledge of mathematics and the use of spreadsheet programs, it is possible to achieve a quality decision with a well-chosen method [26,27].

This method not only supports and qualifies the decisions but also enables the decision-makers to justify their selection and simulates possible results. As we need the most accurate and possible decision, it needs to be consistent and coherent with organizational results [28,29]. It implies that this method can be utilized in the cases/situations/occasions where the management or any individual needs to be involved in the decision-making process, which needs to be supported by the adequate results. The AHP method can be used in personal, logistic, and economic departments. Interestingly, the AHP method can also be used to determine the degree of cave damage. In this case, the AHP method helps to identify major deteriorations or judge conservation orders, etc. [30].

Another use of this method is in agriculture, where the authors evaluated four criteria including "agronomic", "technical", "economic" and "environmental". The AHP method is very useful in all areas. Hence, it is important to know which method of calculation is accurate in order to set the right order of criteria [31]. Another study reported that the inconsistency in judgment and hence in the pairwise comparison matrix of the AHP is the most significant issue to be addressed. Furthermore, analytical hierarchy processing has been widely applied in various case studies and numerous applications [32].

The authors state that cost-benefit analysis and AHP are utilized by the government and public administration for appraising competing alternatives with positive and negative social implications. The AHP is resistant to rank reversal between ratio and difference methods of aggregation. Authors such as Harkar and Vargas argued that the AHP is based on a sound theoretical foundation and is useful for diverse decision-making scenarios [33]. According to Chai and other authors, the AHP method enables the assigning of a value representing the preference degree for a given alternative to each additional alternative [34]. Gupta, Jadhav and Sonar noted that the AHP is the most widely used method for software evaluation [35,36]. Bolpur used the SWOT-AHP-Fuzzy AHP model for formulation and prioritization of ecotourism strategies. The AHP method can be used in combination with other methods [37]. The AHP provides a structured way to analyze complex decision problems and deal with tangible and non-tangible criteria. On the other hand, the AHP provides practical tools for calculating criteria weights and ranking failure modes [38]. The AHP has shown advantages for decision-making when the factors are difficult to measure. Additionally, the AHP is a valid social science research method and is extensively used not only in business management decision-making processes, but also in various areas of information systems research as well [39]. Daengsi, Sirawongphatsara and Pornpongtechavanich used the AHP as an easier decision-making technique to be used to evaluate the considered criteria [40]. Amandeep, Mohammad and Yadav stated that the AHP is a structured technique for dealing with complex decisions based on mathematics and psychology. Furthermore, the AHP gives a complex framework for structuring a decision problem from different areas [1]. Guimarães, Leal and Mendes have applied this new AHP approach in their article [40]. Sakhardande and Gaonkar reported that consistency in pairwise comparisons has been a major hurdle in solving large matrices [41]. The AHP process is a systematic method that simplifies complex problems establishing a hierarchical structure between factors [42]. Ishizaka and Lustis outlined that a high level of agreement between the different scaling techniques and the number of ranking contradictions in-

creases with the dimension of the matrix and the inconsistencies [43]. The AHP has also been applied to supplier and vendor selection, according to Tam and Tummala. Despite its broad applicability, the AHP method suffers from a notable drawback: it requires many comparisons to make a decision [44,45].

Ref. [46] proposed a similar controversial question for Saaty's method relating to its behaviour when an indifferent criterion is added.

By [47–49], the AHP is more suitable for the evaluation of action alternatives considering qualitative criteria rather than quantitative ones.

The selection of the sub-evaluation criteria should be:

Comprehensive: it should include all relevant aspects of the problem. (That is, based on the results and the chosen methods, it is possible to compare the results and determine the ranking of the methods.)

Effective: so that it can be used appropriately in the analysis of the problem (the criteria are used to compare the results).

Decomposable: so that the tasks under investigation can be divided into subtasks.

Not redundant: it does not duplicate some aspects (use only one matrix to calculate).

Minimal: the dimension of the problem should remain as minimal as possible (the number of criteria is sufficient to achieve the results and is also based on the repeated calculation of matrix (B)).

Hypothetical situation (1): the HR manager is making a shortlisting decision. All candidates have equally good results, but the candidates differ in English level, starting salary, years of experience, etc. The HR manager and the area manager cannot decide which candidate is the best fit, so they use the AHP method. They set individual weights for language skills (listening, writing), years of experience, and starting salary. Based on the calculation and the result of the multi-criteria process, the managers can make a better decision and support their decision to the director of the company.

Hypothetical situation (2): The manager is given a number of criteria on the basis of which he has to improve the "image of the company in the field of renewability". The manager has several solutions on the table that can improve the company's renewability, but only one project can be financed. In this case, he can use the AHP method to select the best project.

The key element in both hypothetical cases is the determination of the criteria and the assignment of the individual weights in the criteria. It is advisable to use multiple experts to achieve optimal weights for the criteria. The consistency index is important in the determination of the matrices so that the adjudicator obtains good quality results. The literature states that the consistency index determined should be less than 0.1–10%. In managerial decision making, it is necessary to monitor the consistency index and re-evaluate the weights of the alternatives if the value is higher than 0.1–10%.

Comparison with other methods such as TOPSIS; DEMATEL, ELECTRE etc. can be undertaken in further research.

## 5. Conclusions

The decision-making process is considered as one of the key aspects for entrepreneurs and managers. For quality decision-making, it is necessary to choose an appropriate method which can be involved in decision-making, and it has to be accurate due to its significant effects on the functioning of the organization.

Therefore, the accuracy of the calculations in the managerial decision-making process have to be as precise as possible. The paper compares several methods based on two matrices. From the calculation results, we can detect the same result as for the matrix (A), implying that even when repeatedly calculated with a different matrix, the results are equal. The values of matrix B with the Saaty method: where (A = 0.52081 B = 0.21760 C = 0.12776 D = 0.06675 E = 0.06704) $\lambda_{max}$ = 5.39447 CI = 0.08150.

The results of these calculations are very close and have negligible deviations. Due to the findings of the research, the most accurate method according to the determination

criteria the Saaty method was identified as the most accurate method and subsequently the method of geometric mean.

In conclusion, it is important to emphasize that decision-making via the AHP process in many fields of use, e.g., in companies the criteria can be the size of the company, the number of trucks, and the use of Just in Time, etc. The AHP can be used wherever a high-quality decision based on criteria is necessary. The method must be combined with human decision.

This paper defines the current research problems addressed by the AHP method. Moreover, the AHP method is often used in combination with other methods such as fuzzy, SWOT, etc. Therefore, the acquaintance of this method and several variants of its calculations form a good basis for its expansion.

There were some limitations in the research, such as the limited number of experts to determine the individual weights for the selected areas, but this depends on the selected area. The paper contains the procedure of using the Saaty method through the Excel program.

According to managers, if the result of the Saaty method achieves the correct calculation and ranking of the specific method, the selected method of calculation is not important.

In the second question, 44% of respondents answered that they are capable of using a program to calculate the AHP. There are several software products, but their main limitation is price or subscription. Forty-six percent of respondents said they had experience with some method related to the managerial decision-making process. Seventy-two percent of the respondents think it is important to know and manage some method for decision-making in managerial positions. From the survey results, we can conclude that the outcomes confirmed our assumptions when using the specific calculating methods. Certain methods are simpler, and this is especially appreciated by managers who do not have experience with more complex types of calculations or software.

This article answers the following questions: Are AHP methods differently challenging? What areas is the AHP method used in? What other method is AHP frequently used? Which method of the specified calculations is the most accurate? What limitations occur when using AHP? Is it possible to use Excel to calculate the AHP? Is it important to know and control/understand the method of decision making in management positions among the management staff?

Moreover, further improvements are found in this research over what has been already published so far that can be shown as the salient features and new findings of the paper, as follows:

- The paper compares several methods on the basis of two matrices. The results of these calculations are the same and have negligible deviations.
- This paper defines the current research problems addressed by the AHP method. Furthermore, the AHP method is often used with other methods such as Fuzzy, SWOT, etc. Therefore, the knowledge of this method and several variants of its calculations is a good basis for extending it.
- There were some limitations in writing the paper, such as the limited number of experts to determine the individual weights for the selected areas, but this depends on the area being addressed or the situation with the COVID-19 pandemic, which has caused limited meetings and health scares.
- The article contains the procedure of using Saaty's method through the Excel program. In other words, the student or manager is not forced to pay for the software if Excel is available to him as part of his studies.
- One of the survey findings is that managers believe it is important to know and understand when making decisions. In addition, it would be useful to raise awareness of the usefulness of methods in the decision making process because some managers do not have experience with any method for decision-making.

The paper examines various areas ranging from the accuracy of calculations to managers' opinion of the AHP method.

Table 13 Research objectives with others differ or their method of approach to calculation. Therefore, the table only highlighted the research objectives of the article by another article.

**Table 13.** Less comparison of research with other.

| This Study | Others Studies |
|---|---|
| Defining the AHP method in different research areas with different methods. (2022 researches) | - |
| Determining the accuracy of the AHP method with different calculation methods. Determination of the precession on two matrices. | Determination of the accuracy of the calculation. Use of one matrix. |
| Calculation of the AHP method using Excel. | Make use of other calculation methods. |
| Procedure for all calculations. | - |
| Survey of managers on the decision-making process (in Slovakia) | - |
| - | Use of AHP with another method |

Further research could relate to comparing certain aspects of difficulty with another country, as the short pre-survey is only being carried out in Slovakia. Estonia, Korea, etc. could be chosen for comparison. As already mentioned, another study could be a comparison of AHP with another method from the field of decision-making.

**Author Contributions:** Conceptualization, M.K., J.S., P.M. and K.R.S.; methodology, M.K., J.S., P.M. and K.R.S.; software, M.K.; validation, J.S., P.M. and K.R.S.; formal analysis, J.S. and P.M.; investigation, M.K.; resources, M.K., J.S., P.M. and K.R.S.; data curation, M.K.; writing—original draft preparation, M.K., J.S., P.M. and K.R.S.; writing—review and editing, M.K., J.S., P.M., K.R.S. and V.B.; visualization, M.K.; supervision, M.K., J.S., P.M., V.B. and K.R.S.; project administration, V.B. and P.M. All authors have read and agreed to the published version of the manuscript.

**Funding:** This paper was supported by project VEGA 1/0460/22, Kega 052ŽU-4/2021, KEGA 019TUKE-4/2022, VEGA 1/0524/22 and VEGA 1/0248/21.

**Institutional Review Board Statement:** Not applicable.

**Informed Consent Statement:** Informed consent was obtained from all subjects involved in the study.

**Data Availability Statement:** The data presented are available on request from the corresponding author.

**Conflicts of Interest:** The authors declare that they have no conflict of interest.

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
