# Peer review of "Use of the Analytic Hierarchy Process and Selected Methods in the Managerial Decision-Making Process in the Context of Sustainable Development"

_sustainability, doi:10.3390/su141811546_

Round 1
Reviewer 1 Report (Previous Reviewer 1)
Although I still do not think AHP is a very appropriate decision method in this study, it is clear that the authors worked very hard to revise their manuscript. Therefore, if the authors can carefully revise this manuscript based on other reviewers' comments, then I agree to accept it.
Author Response
Dear reviewer, thank you very much. We are very grateful for your kind and polite appreciation of our work. Also thank you for your recommendation and kind suggestion for other approaches in our next studies. Thank you for your arduous work and instructive advice. Thank you for your comment, that you recommend accepting our manuscript.
Reviewer 2 Report (Previous Reviewer 3)
Reviewer’s comments to authors:
Title of the manuscript - “Use of the Analytic Hierarchy Process and selected methods in the managerial decision-making process in the context of sustainable development"
I have reviewed the revised version of the manuscript entitled “Use of the Analytic Hierarchy Process and selected methods in the managerial decision-making process in the context of sustainable development”. The authors showed how AHP method can be used through different techniques. In the article the authors included methods that can be used in order to calculate the matrix in the AHP process for setting criteria such as Geometric mean, Arithmetic mean, Row sum of the adjusted the Saaty matrix, Reverse sums of the Saaty matrix columns and Row sums of the Saaty matrix. This study also focused on accuracy of various methods used to compute AHP. Indeed there are actually few articles devoted to understanding such a problem. Thus, the article presents an interesting topic.
As per the reviewer comments, the authors revised their manuscript. I feel this paper is suitable for publication in your esteemed journal. Therefore, I accept this manuscript.
Thank you.
Author Response
Dear reviewer, thank you very much. We are very grateful for your kind and polite appreciation of our work. Thank you for your arduous work and instructive advice. Thank you for your comment, that you recommend accepting our manuscript.
Reviewer 3 Report (New Reviewer)
Dear authors,
Thank you for submitting your paper to the Sustainability journal.

Author Response
Dear reviewer,
on behalf of my co-authors, we thank you very much for giving us an opportunity to revise our manuscript, we appreciate you very much for your positive and constructive comments and suggestions on our manuscript entitled “Use of the Analytic Hierarchy Process and selected methods in the managerial decision-making process in the context of sustainable development” (ID: sustainability-1861094). We would like to express our great appreciation to you for your comments on our paper.

This manuscript is a resubmission of an earlier submission. The following is a list of the peer review reports and author responses from that submission.
Round 1
Reviewer 1 Report
Your workload is very substantial, you have done a good job, which is very praiseworthy.
But I have to say that AHP is a more subjective approach to management decision making, and I think you might want to try other approaches. Of course, you can revise your manuscript according to the comments of other reviewers.
Reviewer 2 Report
The purpose of the paper is a bit ambiguous and moreover severity of the problem is not discussed.
- It seems that the Research question is driven by the method rather than the other way round.
- It appears in the beginning that the authors are going to compare AHP (Saaty’s MCDM approach) with other decision-making techniques such as TOPSIS; DEMATEL; ELECTRE etc but the manuscript talks about different approaches that Saaty suggested for calculating Relative weights.
- Also while discussing the approach, managerial implications of terms such as Consistency Index, λmax, and Random Index are not discussed.
- Composite Score calculation for determining ranks can be explained elaborately.
- It seems that in Figure 8 ( Line Number 370-371) some Y-axis labels are missing. For example, the method against 47% is missing.
- Authors claim that Saaty’s method is most appropriate for decision-making in terms of accuracy (although all the approaches discussed have almost the same level of accuracy) but according to Table 12, 69% of respondents disagree with the same. Please Justify
I recommend authors use a simple hypothetical case to explain various approaches of AHP for a better understanding of Management Professionals.
Reviewer 3 Report
Reviewer’s comments to authors:
Title of the manuscript - “Use of the Analytic Hierarchy Process and selected methods in the managerial decision-making process in the context of sustainable development"
I have reviewed the manuscript entitled “Use of the Analytic Hierarchy Process and selected methods in the managerial decision-making process in the context of sustainable development”. The authors showed how AHP method can be used through different techniques. In the article the authors included methods that can be used in order to calculate the matrix in the AHP process for setting criteria such as Geometric mean, Arithmetic mean, Row sum of the adjusted the Saaty matrix, Reverse sums of the Saaty matrix columns and Row sums of the Saaty matrix. This study also focused on accuracy of various methods used to compute AHP. Indeed there are actually few articles devoted to understanding such a problem. Thus, the article presents an interesting topic.
AHP is a method that is frequently employed in several academic disciplines. This study's context, which is somewhat different, demonstrated which AHP calculation technique is more accurate. Therefore, anyone who wants to apply AHP in their research can cite this publication.
I have no negative comments on this manuscript. The work is interesting from my point of view, well structured, well documented and is accessible even to non-specialists. I consider that it fits very well in the area of interest of the journal.
Thank you.